

# Was *Hupehsuchus* a baleen whale-style filter feeder in the Early Triassic? A re-examination of the evidence

Ryosuke Motani[1], Nicholas D. Pyenson[2] and Da-yong Jiang[3]

[1] Department of Earth and Planetary Sciences, University of California, Davis, Davis, California, United States
[2] Department of Paleobiology, National Museum of Natural History, Smithsonian Institution, Washington, District of Columbia, United States
[3] Department of Geology and Geological Museum, School of Earth and Space Sciences, Peking University, Beijing, China

Corresponding author
Nicholas D. Pyenson,
pyensonn@si.edu

## ABSTRACT

One of the recurring paleobiological questions over the last three decades has been whether there were any filter feeding tetrapods before whales evolved. Recently, a study proposed that a small marine reptile from the Early Triassic, *Hupehsuchus nanchangensis*, filter-fed in a mode similar to living right and bowhead whales (Balaenidae). The case for filter feeding was largely based on perceived similarities in dorsal-view cranial morphology between *Hupehsuchus* and Balaenidae, analyzed through geometric morphometrics of 2D landmarks. Here, we show that this similarity was an artifact of multiple errors, including the use of a dataset of extant cetaceans that does not match the morphology of respective species. Notably, 15 of the cetacean species examined were represented by narrow skulls reminiscent of *Hupehsuchus*; without these unrealistic data points, *Hupehsuchus* has no morphospace overlap with any cetaceans, invalidating the proposed inference for Triassic filter-feeding. We collected a new set of landmarks using the published definitions to see how the result changes when using more accurate data along the original authors' intention. We determined that odontocetes and mysticetes do not overlap in morphospace with *Hupehsuchus*, which plots outside any living cetacean species. We conclude that there is insufficient evidence to suggest that *Hupehsuchus* was a filter feeder, in concordance with: energetic studies suggesting balaenid-style feeding would be unsustainable at the small body sizes of *Hupehsuchus*; the lack of an intraoral space for the baleen; and the long neck and comparatively small head that are unsuitable for continuous ram feeding to filter prey-laden volumes of water. This re-examination of *Hupehsuchus* highlights the challenges for inferring filter-feeding in other extinct tetrapods.

## INTRODUCTION

Filter feeding in baleen whales has fascinated scientists for centuries. The last two decades have seen rapid advances in our understanding of this feeding mode, including a marriage between behavioral data and energetic models (*Goldbogen et al., 2011*, *2012*, *2019*) as well

as the discovery of new fossil evidence for its evolution (*Boessenecker & Fordyce, 2015*; *Fordyce & Marx, 2018*; *Peredo et al., 2018*). Fundamentally, filter-feeding in living mysticetes involves the separation of bulk aggregations of prey from volumes of prey-laden water. In mysticetes, there are several modes (*Goldbogen et al., 2017*): unidirectional ram filter-feeding in balaenids (*e.g.*, right (*Eubalaena* spp.) and bowhead whales (*Balaena mysticetus*)); lunge feeding in balaenopterids (*e.g.*, humpback whales (*Megaptera novaeangliae*) and other rorquals); and a generalized mode that includes a mixture of gulping and suction feeding seen in gray whales (*Eschrichtius robustus*). These modes largely map to each of the traditional family-level taxonomic groupings for mysticetes: filter feeding in balaenids and balaenopterids is among the best studied; it is less well documented in gray whales (Eschrichtiidae), which can feed both on the seafloor and in the water column; and it is essentially undocumented in pygmy right whales (*Caperea marginata*, monotypic in Neobalaenidae) (see *Werth & Crompton, 2023*).

No matter the filter-feeding mode, the key component in the process for cetaceans involves the directional flow of prey-laden water across the baleen plates, which function as the filtering apparatus (*Marshall & Pyenson, 2019*). In balaenid-style feeding, the flow of water is continuously unidirectional in that it enters from the front of the mouth and exits from the postero-lateral end of the mouth opening, and using baleen for cross-flow filtration, separating prey from incoming volumes of water (*Goldbogen et al., 2017*). In contrast, the balaenopterid-style feeding has two-way water flow, where water enters and exits through the front of mouth in a tidal fashion, being filtered as it is pushed out (*Goldbogen et al., 2017*). Water is stored between the entrance and exit in the intermandibular pouch that expands like a balloon. The first phase of feeding is accompanied by the acceleration of the whole body, which forces water into the mouth, hence the description of this process as lunge-feeding. However, lunge-feeding may occur in predators without filter organs, such as pelicans, which are typically not considered filter-feeders. Therefore, lunge-feeding is more inclusive mode of feeding than just balaenopterid-style feeding. Likewise, the balaenid-style is often called skim feeding but we will refer to it as the balaenid-style to avoid confusion with skimming by birds, which does not require a filter.

The two styles of filter feeding categorized in this way have sets of underlying mechanisms, some of which are shared. For example, both styles require the baleen as a filter, together with an enlarged intraoral cavity to accommodate it. A large intraoral cavity also allows processing of a vast amount of prey-laden water (*Goldbogen et al., 2017*). The expanded intraoral space is facilitated in part by deepening the oral cavity through arching the rostrum dorsally (Figs. 1B, 1D), although the degree of arching is stronger in the balaenid- than balaenopterid-style (*Dutoit et al., 2023*) because the baleen is shorter and narrower in the latter, which has a mechanism to filter more water with comparatively smaller baleen plates (*Goldbogen et al., 2017*). In the balaenid-style, static lateral arching of the mandible also contributes to the expansion of the intraoral space (Fig. 1C). The balaenopterid-style also benefits from arching of the mandibular rami (Fig. 1B) but the direction of arching is dynamically adjusted through rolling of the respective rami during the feeding behavior—roll here refers to rotation around the longitudinal axis of the object

as in aircrafts and ships (see also *Lambertsen, Ulrich & Straley, 1995*). The arching is directed laterally during lunging to maximize the volume of water stored intraorally. Analogous widening of the mandible during lunging is also seen in pelicans (*Meyers & Myers, 2005*), although it is based on elastic bending of the mandibular rami rather than rolling of a solid curved bone.

Since at least the 1980s, paleontologists have investigated whether there were mysticete-style filter feeders among other marine tetrapods besides whales, especially for marine reptiles in the Mesozoic era (*Collin & Janis, 1997*). Among the candidates was *Hupehsuchus nanchangensis*, a small ichthyosauromorph marine reptile, for which the possible presence of whale-like baleen was suggested based on inferred similarities in the general construction of the rostrum to baleen whales (*Carroll & Zhi-Ming, 1991*). However, their interpretation of the rostral construction of the species, involving whale-like telescoping, was later disproved with additional specimens (*Motani et al., 2015*; *Fang et al., 2023*). *Carroll & Zhi-Ming (1991)* also discussed weaknesses of their baleen inference, noting the lack of direct evidence and difficulty of continuous suspension feeding with the elongated neck and small head of *H. nanchangensis*. The suggestion of filter feeding in *H. nanchangensis* was countered by *Collin & Janis (1997)*, who reviewed the anatomical constraints of diapsid reptiles and linked it to the absence of suspension feeders among marine reptiles. Most recently, *Fang et al. (2023)* reignited the argument by suggesting that *H. nanchangensis* was a balaenid-style filter feeder, largely based on inferred similarities in the dorsal view morphology of the skull between *H. nanchangensis* and extant Balaenidae. *Fang et al. (2023)* used geometric morphometric analysis of nine 2D landmarks from the dorsal view of the skull to justify their argument quantitatively.

*Fang et al.*'s *(2023)* proposal was surprising because several observations make a filter-feeding mode unlikely for *H. nanchangensis*. First, *H. nanchangensis* lacks the intraoral space to place baleen (Figs. 1E–1J), which is essential for mysticete-style filter feeding. The small intraoral space inside a small skull also suggests that the amount of water that can be filtered relative to body size is limited, unlike in filter-feeding whales (see *Pyenson, Goldbogen & Shadwick, 2013*). Also, energetic models and field data have suggested that filter feeding by baleen whales is more efficient at larger sizes, up to the point where prey availability limits the maximum size of the predator (*Goldbogen et al., 2019*). Toward the lower end of their size distribution, filter feeding becomes too costly compared to the energy gained from patchy food. As *Goldbogen et al. (2019*: fig. 4) show, balaenid-style feeding below a body mass of roughly about a ton or slightly more would consume more energy than it would gain, a finding in line with the body size of the smallest fossil balaenids (*Bisconti, Pellegrino & Carnevale, 2021*). The body mass of *Hupehsuchus nanchangensis* is on the order of kilograms rather than thousands of kilograms, below the known limits for mysticete filter-feeding.

The purpose of the present study is to test the hypothesis of balaenid-style filter feeding in *Hupehsuchus nanchangensis* by reexamining the morphological analysis of *Fang et al. (2023)*. Beyond *a priori* arguments against filter-feeding, this reexamination is also motivated by inaccuracies in the original data that we observed and inconsistent methodological steps in the original study. Specifically, many of the species in the dataset

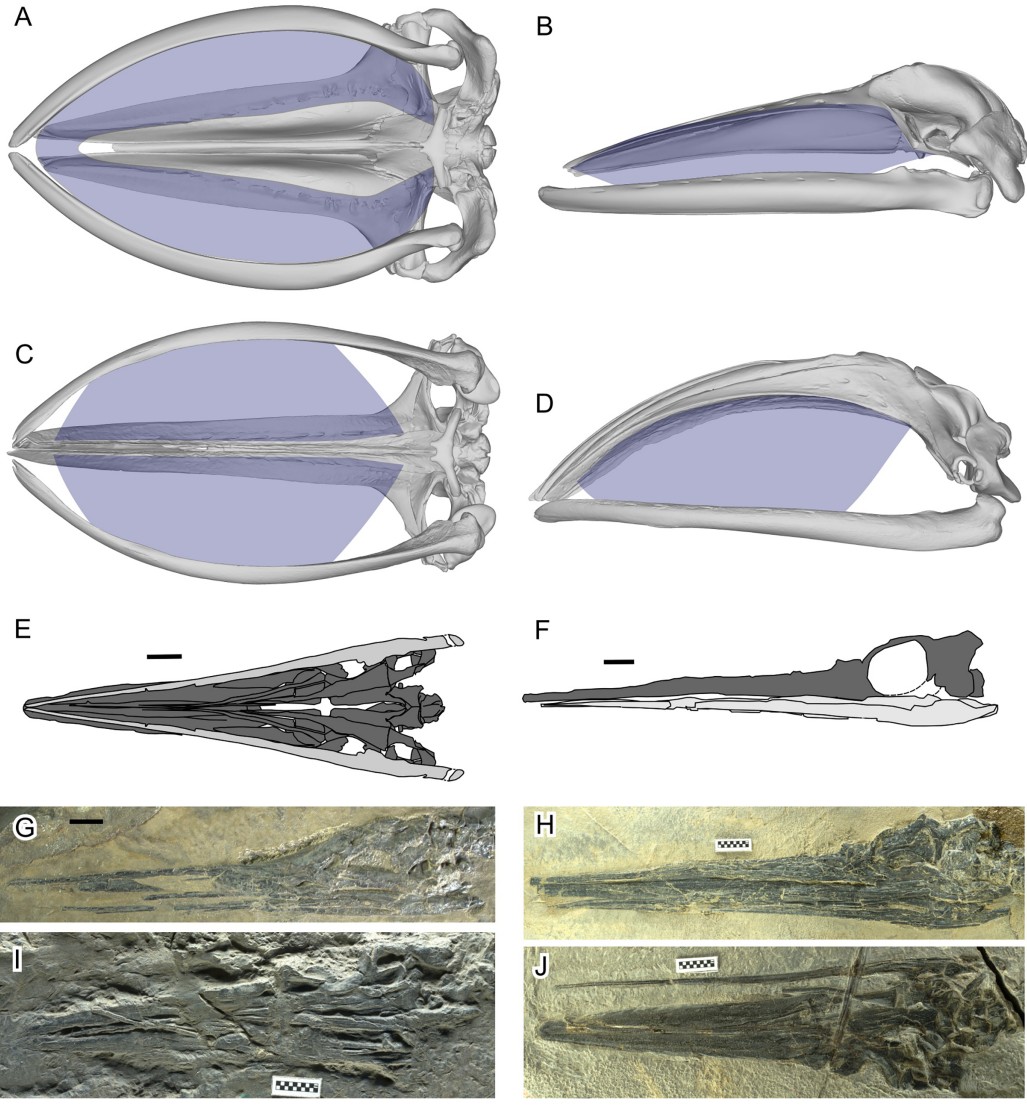

**Figure 1 Filter area for Balaenidae and its absence in *Hupehsuchus nanchangensis*.** (A, B) Humpback whale, *Megaptera novaeangliae*. (C, D) Bowhead whale, *Balaena mysticetus*. (E, F) *H. nanchangensis*. (A, C, E, I, and J) Ventral view; (B, D, F, G, and H) left lateral view. In (E) and (F) light gray is the mandible and dark gray skull. Approximate filter area is colored blue. (A), (B) are based on P12PS00972 at the Glacier Bay National Park. (C), (D) are based on a 3D model of UAM 15988. (E) based on WAGC V26000. (F) based on WAGC V26004. (G) IVPP V3232 (holotype). (H) WAGC V26004. (I) IVPP V4068 (original referred specimen). (J) WAGC V26000. Scale bars are 10 mm.

have landmarks that do not fit the morphology of the species concerned, some landmark definitions are non-homologous, and the original taxon sample contains at least one non-existent species. Also, the landmarks used are not directly relevant to the mechanism of filter feeding—the lateral view of the skull or dorso-ventral view of the mandible would have been a better choice than the dorsal view of the skull as evident from Figs. 1A–1D—so it is doubtful that the published morphospace can distinguish among feeding styles. Therefore, we tested whether the landmark data accurately reflect the morphology of the

species concerned, and whether the published morphospace can distinguish among feeding styles. We also tested whether the published analytical results are reproducible when: (1) using a new set of landmarks placed according to the published definitions; (2) using homologous landmarks; and (3) using 3D rather than 2D landmarks.

## MATERIALS AND METHODS

Our analysis took the following steps. First, we scrutinized the data for accuracy of landmarks as well as validity of taxon selection. Second, we verified the repeatability of the published analysis by *Fang et al. (2023)* to make sure that the published data gave rise to published results. Third, we tested whether the published analysis is relevant to the feeding style of *Hupehsuchus* by adding a non-filter-feeding ichthyosauromorph to the data. Fourth, we tested how removal of inappropriate data identified in our second step affected the analysis. Fifth, we collected our own landmarks based on the definitions given by *Fang et al. (2023)* and compared the results with the published study. Finally, four of the nine landmarks used by *Fang et al. (2023)* had non-homologous definitions that made them jump across a few anatomical positions depending on the species, so we tried to convert them to a homologous set that accounts for all possibilities of the anatomical positions of the four. We then analyzed the homologous set of landmarks for comparison with the original study. We experimented with both 2D and 3D coordinates in this last step.

### Filter area

As already presented in Fig. 1, filter areas were compared between bowhead whales (*Balaena mysticetus*) and *Hupehsuchus nanchangensis*. For *B. mysticetus*, filter area is based on *Dutoit et al. (2023)*, with baleen extent based on *George et al. (2016)*. For *H. nanchangensis*, the lateral and ventral views of the skull were approximately reconstructed based on WGSC V26004 (Fig. 1H) and V26000 (Fig. 1J), respectively. The reconstructed images were cross-checked against the holotype (IVPP V3232, Fig. 1G) and an original referred specimen (IVPP V4068, Fig. 1I), which expose the lateral and ventral views, respectively, although the preservation is not as good as in the two WGSC specimens.

### Assessment of the published taxon list

We examined the list of cetacean taxa in the dataset of *Fang et al. (2023)* for taxonomic validity and suitability for landmarking. For example, two of the nine landmarks used by *Fang et al. (2023)* require a pair of nasals to be present, but not all cetaceans have both of these bones.

### Accuracy of published landmarks

Landmarks used by *Fang et al. (2023)* were obtained from their tables S1 and 2 in Supplemental Information. We downloaded it three times on different days to make sure that there were no accidental modifications during downloading. The dataset was imported into R and the landmarks were plotted as dots and lines connecting them per species in a PDF file, making sure that the aspect ratio of the plot is kept at isometry, *i.e.*, x and y axis have the same scale so that the resulting polygons have natural proportions

between the skull length and width. The plots were compared with orthographic projections of the skull of corresponding species from 3D models. They were also compared with similar plots from our own landmark sets.

### *Hupehsuchus* occiput

The skull reconstruction of *Hupehsuchus* by *Fang et al. (2023*: fig. 2), which they used for landmarking purposes, ignored the occiput although they figured semi-articulated occipital bones posterior to the skull roof for the new specimen that they reported (2020-NYF-84-4 in their fig. 3). Using the occiput would affect the choice of the most posterior points of the skull, which are landmarks 3 and 5 in the original study, as well as the most posterior point along the sagittal plane (their landmark 4). We therefore added the occiput to their fig. 2 based on their fig. 3, as given in Fig. S1 here.

The preserved position of the basioccipital relative to the skull roof depends on the specimen to some extent. For example, the laterally exposed skull of WGSC V26004 suggests that the basioccipital is slightly anterior to the posterior end of the supratemporal-squamosal complex (Fig. 1H), whereas the ventrally exposed skull of WGSC V26000 reveals the basioccipital slightly behind the supratemporal (Fig. 1J) although not as far back as in 2020-NYF-84-4. We therefore took the middle ground and placed it at about the level of the posterior end of the supratemporal-squamosal complex. Figure S1 was made with the following steps. First, the outlines of the occipital bones were traced from the published figure (*Fang et al., 2023*: Fig. 3). Second, the bones were reidentified—what was labeled the exoccipital is more likely the opisthotic given its similarities to the opisthotic in *Chaohusaurus*. Third, the bones were rearranged to their presumable anatomical positions. Fourth, supraoccipital and opisthotic were antero-posteriorly shortened to account for their inclination against the horizontal plane, to the degree the posterior end of the basioccipital is leveled with that of the supratemporal. The basioccipital and basisphenoid are exposed to show their horizontal planes so they were not shortened.

### 3D model preparation

3D surface mesh models of crania were downloaded for 64 species of cetaceans spanning 10 families and 33 genera, from three online sources: Morphosource (morphosource.org), Phenome10K (phenome10K.org), and Sketchfab (sketchfab.com). The list of species and URL for each model is given in Table S1.

The skull models were variously oriented in the downloaded models, so they were first aligned to their principal axes and then orthogonal rotations were applied to make the bilateral, dorsoventral, and antero-posterior directions match X, Y, and Z axes, respectively. This was necessary for consistent landmarking of the most posterior and lateral points as explained below. A small number of 3D models had extreme asymmetry that led to a misalignment between anatomical axes and geometric principal axes based on surface polygons. These cases were treated manually in Meshlab (*Cignoni et al., 2008*) with minor adjustments of the yaw angle and sometimes the roll angle but never the pitch angle.

## New landmarks

### *Cetaceans*

The aligned models from the previous process were imported into 3D Slicer (*Kikinis, Pieper & Vosburgh, 2014*) for 3D landmark placement. The original landmark definitions of *Fang et al. (2023)* contained the most posterior points and widest points of the skull, which may change slightly depending on the orientation of the skull relative to the view, as well as field of view (FOV), as demonstrated in Fig. 2. When using photographs, the most posterior point of the skull may vary between the dorsal and ventral views (*e.g.*, Figs. 2C *vs.* 2D). We therefore made sure to place landmarks under orthographic projection in strictly dorsal or ventral views according to principal axes of the model. The skull outline in such a view matches exactly that from the ventral view (Figs. 2A *vs.* 2B) so its use removes the above-mentioned bias.

We first used the landmark definition of *Fang et al. (2023)* to reproduce the original study. We needed to interpret the descriptions for landmarks 8 and 9, which are "Anterormedial" and "Posteromedial point of nasals", respectively (table 1 of *Fang et al., 2023*). The definitions do not allow us to pick a point in a repeatable fashion. Their fig. 2 placed the two landmarks at the anterior and posterior ends of the internasal suture, respectively, so we used those repeatable definitions.

Some of the landmarks as defined by *Fang et al. (2023)* suffer from the lack of homology, which is against the basic principles of landmarks in geometric morphometrics. For example, the most posterior point of the skull may be found on the paroccipital process as in minke whales (*Balaenoptera acutorostrata*, Fig. 3G), occipital condyles as in bowhead whale (*Balaena mysticetus*, Fig. 3C) and most odontocetes, or on the squamosal as in *B. mysticetus* viewed differently ventrally through a camera lens (Fig. 2). Treating them as the same morphological feature is problematic since there is no anatomical consistency. A simple solution is to landmark all three features separately in this case—six in total because they are paired. Similarly, the widest point of the skull may be on the frontal or squamosal, so they should be treated separately. These newly designated landmarks are still not Type-1 landmarks but at least of Type 2 or 3, whereas the respective original landmarks do not qualify as proper landmarks. The list of the expanded set of landmarks and their definition are given in Table 1 and example landmark placements are figured for three species in Fig. 3.

### *Fossil taxa*

Two fossil taxa were included in the analysis, *Hupehsuchus nanchangensis* and *Chaohusaurus brevifemoralis*—see below for why this latter species was added. For these taxa, specimens have been compacted during preservation, distorting their 3D morphology at least to some extent. While it is impossible to find the exact 3D placement of the landmarks in life for these taxa, reasonable approximations may be made by combining the lateral and dorsal views of the skull. We therefore placed the 9 and 15 landmarks explained above in each of the dorsal and lateral views of the skull in respective species. We then combined the antero-posterior and bilateral coordinates from the dorsal view and dorso-ventral coordinates from the lateral view. *C. brevifemoralis* was landmarked based

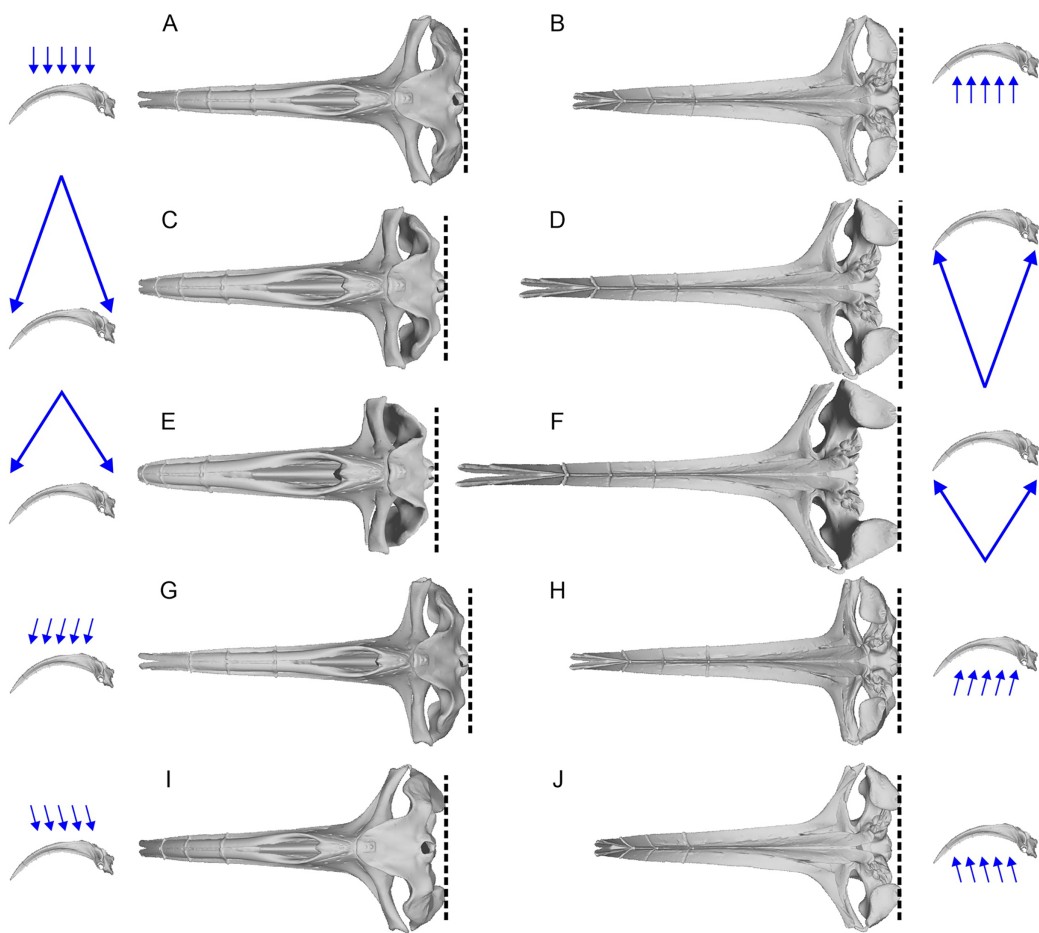

**Figure 2 Variation of the most posterior point of the skull in bowhead whales (*Balaena mysticetus*) depend on the viewing direction and field of view (FOV).** The lateral images of the skull with blue arrows show the direction and FOV of their associated dorsoventral images. Broken lines mark the posterior end of the skull in each view. (A, B) Orthographic projection from the strictly dorsal or ventral direction, which is the preferred setting in this study; (C, D) FOV of a typical 50 mm lens for film cameras from the strictly dorsal or ventral direction; (E, F) FOV of a typical 35 mm lens for film cameras from the strictly dorsal or ventral direction; (G–J) orthographic projection from angles 10 degrees tilted in pitch from the strictly dorsal or ventral direction. The most posterior points of the skull are on the occipital condyle, but it may appear as if they are on the squamosal depending on the direction and FOV, as in some landmarks of *Fang et al. (2023)*. Based on a 3D model of NHMUK 1986.116.

on a pair of photographs from the holotype given in Fig. S2, whereas the dorsal view reconstruction by *Fang et al. (2023)*, as revised in Fig. S1 as explained above, was combined with the lateral view of WGSC V26004 for *H. nanchangensis*, as in Fig. S3.

Both *Hupehsuchus nanchangensis* and *Chaohusaurus brevifemoralis* have some uncertainties in terms of the relative position of the occipital condyle to the supratemporal-squamosal complex antero-posteriorly because of the known variations in observed placement. We placed them at the same level to be neutral, given that either one may be slightly more anterior or posterior in preservation. However, the landmark definition of *Fang et al. (2023)* forces us to pick one of them as more posterior to the other.

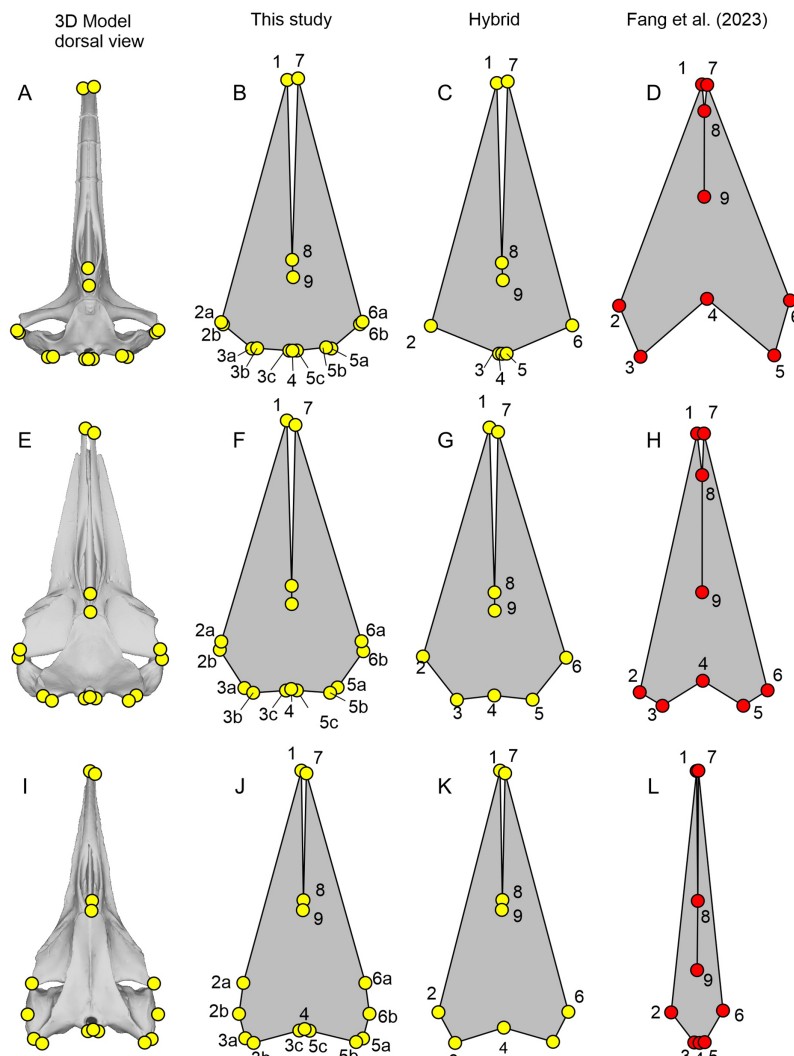

**Figure 3 Landmark placement in this study and *Fang et al. (2023)*.** (A–D) *Balaena mysticetus* (NHMUK 1986.116). (E–H) *Balaenoptera acutorostrata* (NHMUK 1965.11.2.1). (I–L) *Caperea marginata* (NHMUK 1876.2.16.1). (A), (E), and (H) depict how the 15 homologous landmarks fitted to the skull. (B), (F), and (J) are polygons from the 15 homologous landmarks of this study. (C), (G), and (K) are nine landmarks we reproduced based on the definition of *Fang et al. (2023)*. (D), (H), and (L) are the same nine landmarks, plotted from the published data of *Fang et al. (2023)*. Yellow landmarks were placed by this study, while red landmarks are from *Fang et al. (2023)*. Notice the large discrepancy between red and yellow landmarks.

We coped with this by including both possibilities. That is, for each species, we have a set of landmarks where the most posterior points of the skull are placed on the basioccipital, and another set where the points are placed on the squamosal-supratemporal complex. This step is only relevant when using the nine landmarks defined by *Fang et al. (2023)* because there is no need to force one of them to be more posterior than the other when using the new 15 homologous landmarks.

**Table 1 Definitions of the 15 landmarks used in this study.**

| Fig. 3 | Serial | Definition |
|---|---|---|
| 1 | 1 | Anterior tip of the left premaxilla |
| 2a | 2 | Most leftward point of the left frontal |
| 2b | 3 | Most leftward point of the left squamosal/supratemporal |
| 3a | 4 | Most posterior point of the left squamosal/supratemporal |
| 3b | 5 | Most posterior point of the left paroccipital process |
| 3c | 6 | Most posterior point of the left occipital condyle |
| 4 | 7 | Most posterior point along the midline of the occipital condyle |
| 5c | 8 | Most posterior point of the right occipital condyle |
| 5b | 9 | Most posterior point of the right paroccipital process |
| 5a | 10 | Most posterior point of the right squamosal/supratemporal |
| 6b | 11 | Most rightward point of the right squamosal/supratemporal |
| 6a | 12 | Most rightward point of the right frontal |
| 7 | 13 | Anterior tip of the right premaxilla |
| 8 | 14 | Anterior end of the nasals along the midline |
| 9 | 15 | Posterior end of the nasals along the midline |

## GPA and PCA

The landmarks were aligned by generalized Procrustes analysis (GPA) and then analyzed by principal component analysis (PCA) in the geomorph package of R (*Adams & Otarola-Castillo, 2013*).

### Repeatability of published results based on published data

We ran GPA and PCA using the published data set from tables S1 and 2 of *Fang et al. (2023)* and compared the plot of PC1 *vs.* PC2 with fig. 4 of *Fang et al. (2023)*.

### Test of functional significance of published morphospace

To test if the published principal component space has any relevance in feeding mechanics, we added a species that is clearly not filter feeding while being related to *Hupehsuchus nanchangensis*, to see where it appears in morphospace. We chose *Chaohusaurus brevifemoralis* because its skull is well known from multiple specimens (*Huang et al., 2019*), its typical body size is about the same as in *H. nanchangensis*, and it is approximately coeval to *H. nanchangensis*, being from the same late Spathian (Early Triassic). It has a narrow and elongated snout with numerous teeth and without any signs of a filtering structure, suggesting that it is not a filter feeder.

### Removal of problematic taxa

We removed problematic species from the published data of *Fang et al. (2023)* to see their effects on the principal component space. We first removed species that failed the assessment of taxon screening explained above. We then removed those species whose published landmarks are extremely different from the true skull morphology.

### Removal of non-cetacean extant species

We tried removing non-cetacean extant species from the data of *Fang et al. (2023)* to see if it affects the overall distribution of cetacean taxa and *Hupehsuchus nanchangensis* in the resulting principal component space. If the effect is limited, it justifies removal of non-cetacean taxa in the subsequent analyses.

### New datasets

We analyzed the new landmarks that we collected from the 3D models of cetacean cranium, as well as those from orthogonal photographs of *Hupehsuchus nanchangensis* and *Chaohusaurus brevifemoralis* with GPA and PCA. We tried nine landmarks based on the published definitions of *Fang et al. (2023)*, as well as 15 homologous landmarks defined in this study (Table 1). For each set, we tried two variations—first with all of 3D coordinates, and second with only coordinates along the longitudinal and bilateral axes of the skull that are visible dorsally, which are equivalent to the landmarks collected from dorsal-view only.

## RESULTS

### Assessment of published taxon list

We found three of the 67 species of cetaceans in the dataset of *Fang et al. (2023)* to be inappropriate for the analysis. *Kogia breviceps* and *K. sima* lack nasals (*Jefferson, Webber & Pitman, 2015*) and cannot be landmarked for positions 8 and 9 (anterior and posterior ends of the internasal suture) of the nine landmarks—it is unclear how *Fang et al. (2023)* still managed to place these landmarks. Similarly, *Physeter macrocephalus* cannot be landmarked for the same positions because it has only one nasal (*Jefferson, Webber & Pitman, 2015*), thus lacking the internasal suture. These three species were still included in some of our reanalyses for the sake of repeatability testing but we did not include them in our own dataset. This revised taxon list lowers the effective taxonomic diversity of cetaceans in the data of *Fang et al. (2023)* to be nine families, 32 genera, and 63 species, which is only slightly less than in our new dataset (10 families, 33 genera, and 64 species).

### Accuracy of published landmarks

The plots of the landmarks for all species in the dataset of *Fang et al. (2023)* are given in Fig. S4, directly based on the published dataset. A part of these landmarks is reproduced in Figs. 3–5 with comparisons to the actual skull morphology. Comparisons of the new landmarks with the published ones are given in Fig. S5 for all species that are shared between the two datasets. We only compared the cetacean part of the data with the skull morphology of the respective species because it suffices for the purpose of this study, as noted below.

We found that most of the cetacean species had been given landmarks that do not fit their respective cranial morphology. For example, the anterior end of the internasal suture (landmarks 8) is placed too anteriorly in a vast majority of species. Also, the occipital condyle is ignored in some species (*e.g.*, Fig. 3D) while not in others (Fig. 3L), resulting in

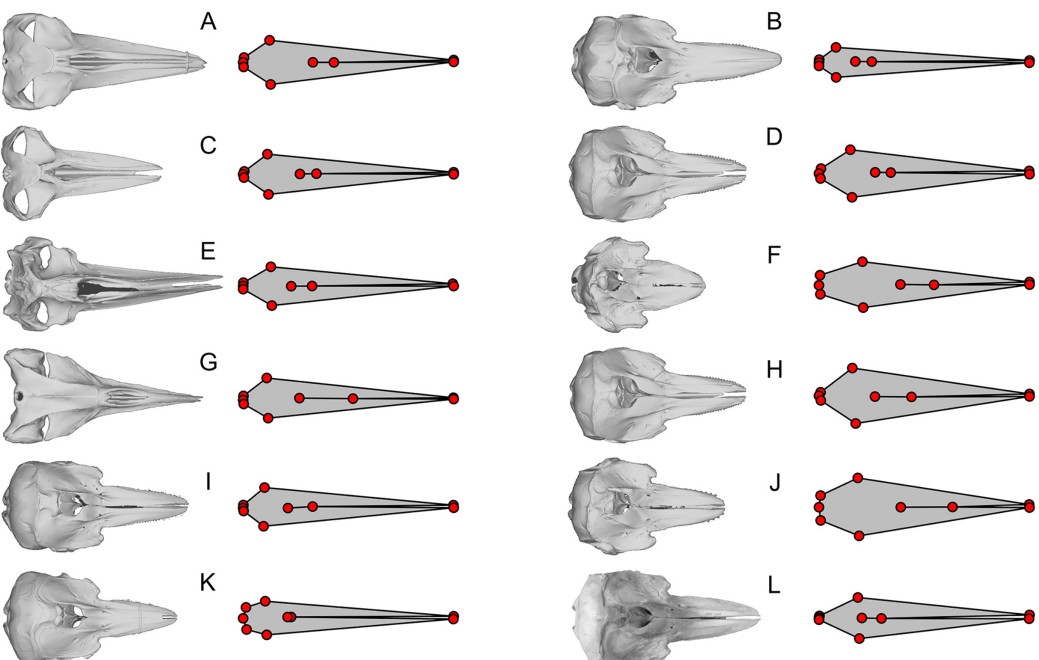

**Figure 4 Examples of extreme discrepancy between skull morphology and published landmarks.** The dorsal view of the skull is compared with the published landmarks of *Fang et al. (2023)*. Notice that the published landmarks for each species suggest much narrower skulls than the actual skulls of the respective species, given left to the landmarks, across mysticetes and odontocetes. These unrealistically narrow skulls confused the analysis to mistake these cetaceans to be similarly shaped to *Hupehsuchus nanchangensis*, which also has a narrow skull. (A) *Balaenoptera borealis* (NHMUK C.1934.5.25.1). (B) *Cephalorhynchus commersoni* (USNM 550156). (C) *Megaptera novaeangliae* (P12PS00972). (D) *Lagenorhynchus albirostris* (AMNH 37162). (E) *Eschrichtius robustus* (USNM 13803). (F) *Globicephala macrorhynchus* (USNM 500239). (G) *Caperea marginata* (NHMUK 1876.2.16.1). (H) *Grampus griseus* (USNM 500271). (I) *Ce. eutropia* (NHMUK 1881.8.17.1). (J) *Feresa attenuata* (USNM 504917). (K) *Ce. heavisidii* (NHMUK 1948.7.27.1). (L) *Ce.* (MNZ MM002607). The skull in (L) is an online photograph from the New Zealand Museum (https://collections.tepapa.govt.nz/object/1030562) and the rest of the skull are orthographic projections from respective 3D models listed in Table S1.

inconsistent placement of the three landmarks along the posterior end of the skull (landmarks 3 to 5) across taxa. The widest points of the skull (landmarks 2 and 6) are often placed too anteriorly or posteriorly—for example, *Balaenoptera acutorostrata*, which *Fang et al. (2023*: fig. 2B) figured as an example of landmark placement, has landmarks 2 and 6 much more posteriorly displaced in the data file (compare Fig. 3H here with their fig. 2B).

  Most problematically, there is a collection of 15 cetacean species that has been given landmarks that do not even resemble the true skull morphology—twelve of them are figured here (Figs. 3L, 4). Apart from these twelve, *Delphinus delphis*, *Lagenorhynchus albirostris*, and *L. hosei* are also given similarly arranged landmarks that are narrower than the skull, although slightly wider than those which are figured. In all cases, these landmarks depict unusually narrow skulls for a cetacean that are reminiscent of the proportion in *Hupehsuchus nanchangensis*. The 15 are distributed across three families of mysticetes and one family of odontocetes. These 15 biased the results of *Fang et al. (2023)* as shown below.
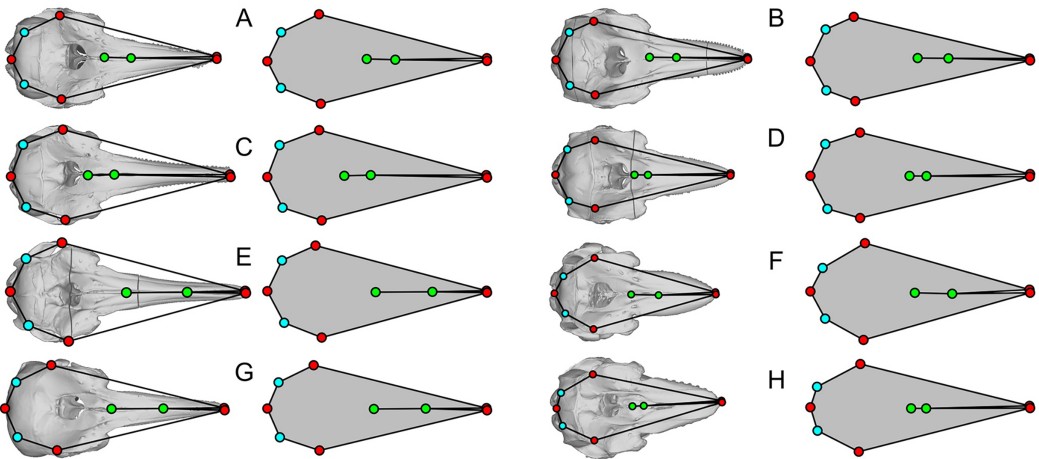

**Figure 5 Examples of shared errors across similarly arranged landmarks.** The dorsal view of the skull is compared with the published landmarks from *Fang et al. (2023)*. Green landmarks should be behind the blowhole in cetaceans but are placed far anteriorly. Light blue landmarks should be placed at the posterior end of the skull but placed more anteriorly. The eight landmark sets are similar but have wrong length/width ratios for at least half of the skulls figured. (A) *Sousa teuszii*. (B) *Lagenorhynchus obliquidens*. (C) *Sou. sahulensis*. (D) *L. australis*. (E) *Sou. plumbea*. (F) *Orcinus orca*. (G) *Sotalia* sp. (H) *Pseudorca crassidens*.

Apart from these 15, there is another collection of eight cetacean species that stand out for having landmarks that closely resemble each other while sharing common errors of misplacing four of the landmarks in the same way (Fig. 5)—they all have the nasal landmarks displaced anteriorly when they are supposed to be behind the external bony naris (Fig. 5, green dots), whereas the most posterior points of the skull are misplaced antero-laterally (Fig. 5, light blue dots). In at least half of the species, the landmarks draw a shape that is too narrow for the skull (Figs. 5B, 5D, 5F, and 5H). There are more isolated examples of discrepancy between the published landmarks and skull morphology that we did not mention. However, the examples given here are sufficient to establish the low accuracy of the published dataset.

## GPA and PCA

### Repeatability of published results based on published data

The result is given in a principal component space in Fig. 6A, which is identical to fig. 4A of *Fang et al. (2023)* when accounting for differences in color and aspect ratio of the graph. Therefore, the published dataset can reproduce the published results using the published methods. However, when the inappropriate data are removed, the original results do not hold, as elaborated below.

### Test of functional significance of published morphospace

The result is given in Fig. 6B. The addition of the occiput moves *Hupehsuchus nanchangensis* slightly closer to Balaenidae, although the difference is trivial because *H. nanchangensis* remains at the other end of the mysticete distribution from Balaenidae (*i.e.*, it is closer to Balaenopteridae than Balaenidae). *Chaohusaurus brevifemoralis* appears
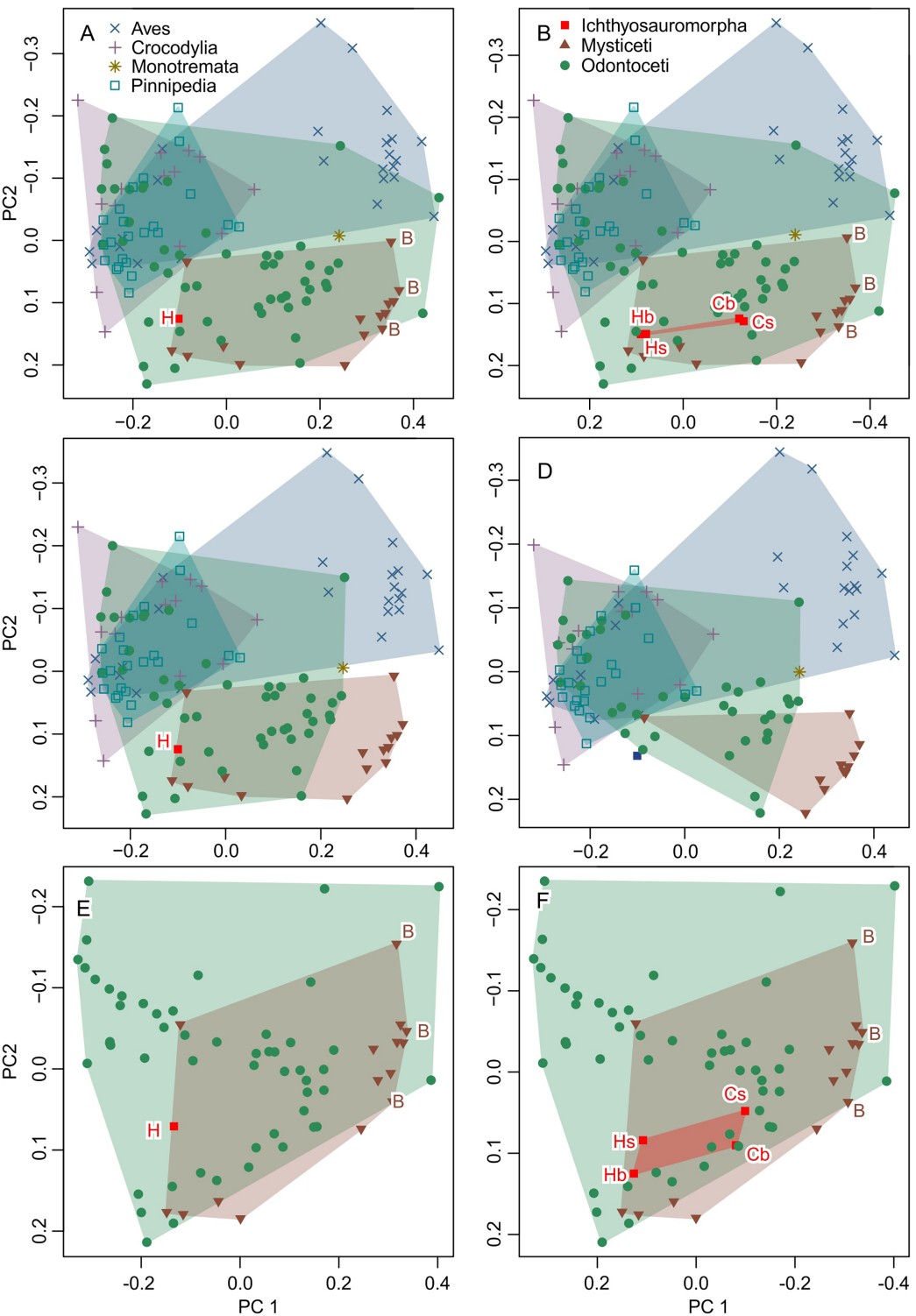

**Figure 6 Partial principal component space based on the data from _Fang et al. (2023)_.** (A) Reanalysis without any modification to test the repeatability of the analysis. (B) Same as (A) but the data for Ichthyosauromorpha have been revised though additions of _Chaohusaurus brevifemoralis_ and the occiput of _Hupehsuchus nanchangensis_ (see Methods). (C) Same as (A) but four species that are invalid or lack bones necessary for landmarking are removed. (D) Same as (C) but 15 species with landmarks that are far too narrow (_e.g._, Fig. 4) are removed. (E) Same as (A) but with non-cetacean extant species removed.

**Figure 6** (continued)
(F) Same as (B) but with non-cetacean extant species removed. Symbols: B, Balaenidae; b, basioccipital considered most posterior; C, *C. brevifemoralis*; H, *H. nanchangensis*; s, squamosal-supratemporal complex considered most posterior.

closer to Balaenidae than *H. nanchangensis*, being located near the center of the mysticete distribution. Given that *Chaohusaurus* is not a filter-feeder, its position near the center of mysticetes suggests that the morphospace of dorsal views of tetrapod crania alone cannot establish the feeding styles of extinct taxa.

### Removal of problematic taxa

When removing three species that lack the internasal suture, the distribution of odontocetes shrinks in the resulting principal component space along PC1 but the effect is limited otherwise (Fig. 6C). Removal of 15 more species with extremely unreasonable landmarks, the distributions of both odontocetes and mysticetes become narrower (Fig. 6D). Notably, *Hupehsuchus nanchangensis* is no longer nested within the cetacean distributions after the removal. Therefore, we argue that placement of *H. nanchangensis* within the cetacean distribution in fig. 4 of *Fang et al. (2023)* is an artifact of having these unrealistic data points.

### Removal of non-cetacean extant taxa

Removal of non-cetacean taxa has minimal effects on the distributions of cetacean taxa, *Hupehsuchus nanchangensis*, and *Chaohusaurus brevifemoralis* in the resulting morphospace (compare Figs. 6E and 6F with Figs. 5A and 5B, respectively). There is a slight difference observed along PC2 but not PC1—this difference is best seen in Ichthyosauromorpha in Fig. 6F. Therefore, major conclusions of *Fang et al. (2023)* can be tested without non-cetacean extant taxa.

### New dataset

New data set results in morphospaces that are almost completely different from that based on the published dataset (compare Figs. 6E, 6F with Fig. 7). When using the best of the four datasets, with 3D coordinates of all 15 landmarks in Table 1, ichthyosauromorphs, mysticetes, and odontocetes are clearly separated from each other with wide gaps in between them (Fig. 7A). PC1 alone can separate ichthyosauromorphs from cetaceans, although mysticetes and odontocetes overlap along this axis. When using only 2D coordinates of the same 15 landmarks, the result does not change very much (Fig. 7E). However, it is no longer possible to separate ichthyosauromorphs from cetaceans based only on PC1 alone, partly because the whole space is slightly rotated counterclockwise compared to the previous case (compare Figs. 7A and 7E)—see Discussion.

When using the 3D coordinates of the nine landmarks as defined by *Fang et al. (2023)*, the results are essentially similar to the two cases with 15 landmarks (compare Figs. 7E with 7A) in terms of taxonomic distribution, although the separations between groups are narrower. However, the nature of PC2 changes substantially—with 15 landmarks, PC1 is largely controlled by cranial height while PC2 is the relative position of the skull roof to the

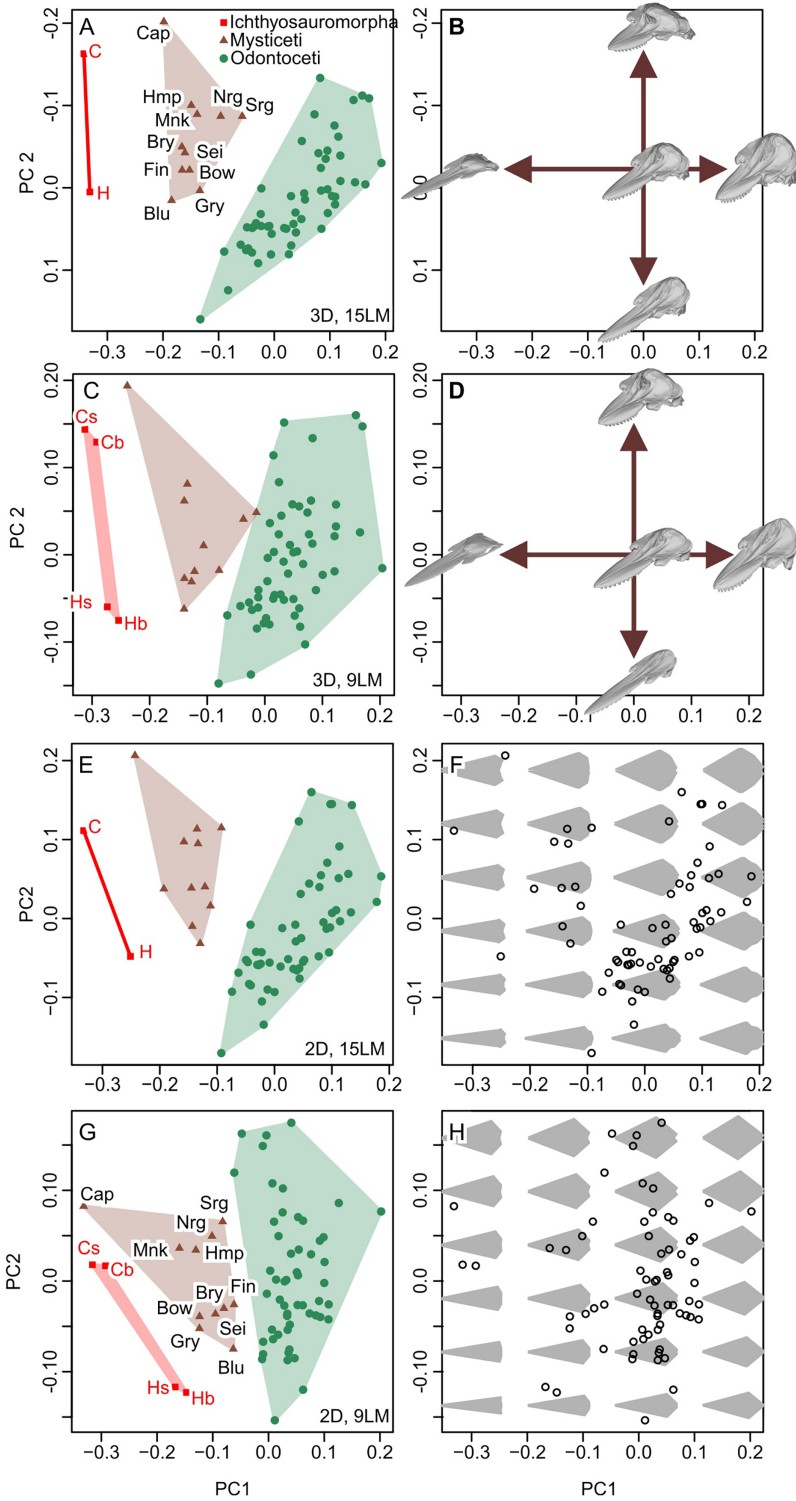

**Figure 7 Partial principal component space based on the new data.** (A, B) Using 3D coordinates of 15 landmarks in Table 1. (C, D) Using 3D coordinates of nine landmarks as defined by *Fang et al. (2023)*. (E, F) Using 2D coordinates of the 15 landmarks in Table 1. (G, H) Using 2D coordinates of nine landmarks as defined by *Fang et al. (2023)*. Symbols: b, C, H, and s as in Fig. 6. Blu, Blue whale (*Balaenoptera musculus*); Bow, Bowhead whale (*Balaena mysticetus*), Bry, Bryde's whale (*Balaenoptera edeni*); Cap, Pygmy right whale (*Caperea marginata*); Fin, Fin whale (*Balaenoptera physalus*); Gry, Gray

**Figure 7** (continued)
whale (*Eschrichtius robustus*); Hmp, Humpback whale (*Megaptera novaeangliae*); Mnk, Minke whale
(*Balaenoptera acutorostrata*); Nrg, Northern Atlantic right whale (*Eubalaena glacialis*); Sei, Sei whale
(*Balaenoptera borealis*); and Srg, Southern right whale (*Eubalaena australis*).

mandibular articulation (Fig. 7B) but this distinction is blurred when using nine landmarks because both PC1 and 2 are largely affected by skull height (Fig. 7D). When using only the 2D coordinates of the same nine landmarks, the pattern is still similar, but all three groups overlap along PC1 (Fig. 7G), which is associated with changes in the nature of PC1 and 2. With 15 landmarks, PC1 is largely controlled by the relative position of the basioccipital to the mandibular articulation (Fig. 7F) but this relative positioning is reflected in both PC1 and 2 when using nine landmarks (Fig. 7H).

## DISCUSSION

There are substantive differences between the results from our new landmarks and those published by *Fang et al. (2023)*, even when we tried to reproduce the original study by using their published definitions of landmarks and only 2D coordinates to match theirs—compare Figs. 6F with 7G. Most notably, our study shows that *Hupehsuchus nanchangensis* is no longer nested within the cetacean distributions, which clearly separate major cetaceans clades apart from each other. In contrast, the results of *Fang et al. (2023)* suggested that the morphological variation of odontocetes almost entirely encompasses that of mysticetes, which contradicts the common view that mysticetes and odontocetes have disparate cranial morphology (*Coombs et al., 2022*).

We think that primary cause of the differences between our results and those of *Fang et al. (2023)* can be attributed to the low accuracy of landmark data. Less than 20% of the cetaceans in the original published data by *Fang et al. (2023)* have landmarks that fit the cranial morphology of the respective species even approximately. Most problematic is a subset of 15 species with unrealistic landmarks (partly depicted in Figs. 3L and 4). Without these 15 species with very narrow skulls reminiscent of *Hupehsuchus nanchangensis*, it is hard to justify a similarity between *H. nanchangensis* and baleen whales as proposed by *Fang et al. (2023)*. We first considered the possibility that these discrepancies may have been caused by a simple taxonomic mislabeling but subsequently abandoned the idea because the number of landmarks that fit odontocetes and mysticetes, respectively, are fewer than the number of species from these clades that are supposed to be present in the data. Likewise, it is perplexing why four species in Figs. 5B, 5D, 5F, and 5H were given landmarks that obviously do not fit the skull morphology while closely resembling Figs. 5A, 5C, 5E and 5G. These observations, among others, cast doubt on the reliability of their published data.

A majority of cetacean species in the data of *Fang et al. (2023)* were landmarked based on small line drawings of the skulls in the list of taxonomic identification keys (*Jefferson, Webber & Pitman, 2015*), which are not intended for accurate landmarking; moreover, the landmarks of *Fang et al. (2023)* differ from these drawings too. It is puzzling as to why they

did not use the lateral view of the skull to augment the dorsal view when *Jefferson, Webber & Pitman (2015)* gave the lateral view. Finally, it would have been both convenient and accurate to use the 3D models of cetacean skulls published by *Coombs et al. (2022)* as a means to consistently anchor any 2D morphometric comparisons.

The results of *Fang et al. (2023)* were also affected slightly by non-homologous landmarks, as well as 2D rather than 3D coordinates, although the effect is trivial compared to the low accuracy of their landmarks. A comparison of the panels of Fig. 7 shows that taxonomic differences become better reflected in PC1 as the informational content of the data improves. Thus, the use of 2D landmarks instead of 3D rotates the principal axes counterclockwise in Fig. 6, lowering the correlation between PC1 and taxonomy. This is an acceptable difference because some of the major differences between the clades are only seen in lateral view that is not accounted for by the 2D datasets. For example, the squamosal-supratemporal complex is placed dorsally in ichthyosauromorphs but ventrally for cetaceans. Similarly, the use of non-homologous landmarks instead of a homologous set narrows the gaps between taxonomic divisions. This result stems from taxonomic differences that become indistinct by treating non-homologous anatomical positions as if they were equivalent (*Zelditch, Swiderski & Sheets, 2012*).

Lastly, the interpretation of the resulting morphospace needs to be made in the context of the mechanisms behind the function in question. No mechanism was considered by *Fang et al. (2023)* while the interpretation of the results was biased toward the preferred conclusion. For example, fig. 4A of *Fang et al. (2023)* shows that *Hupehsuchus nanchangensis* was well-nested inside the odontocete distribution while being located at the edge of mysticete distribution. An objective interpretation of the plot would be that the morphospace cannot distinguish whether the cranial morphology of *H. nanchangensis* is more similar to odontocetes or to mysticetes. Also, their fig. 4B shows that *H. nanchangensis* is well nested inside the distribution of those predators feeding on mid-sized prey while also being located in the distribution of those feeding on tiny food. An objective interpretation of this plot would be that the morphospace cannot distinguish between these two food types. Nevertheless, *Fang et al. (2023)* selected only one of the four possible combinations. Of course, these are now moot points since the plots themselves were most likely artifacts of unreliable data collection, and that it cannot be used to infer the feeding style as demonstrated above.

Apart from the quantitative analysis discussed so far, *Fang et al. (2023)* also gave several qualitative reasons to support their hypothesis that *Hupehsuchus nanchangensis* was a balaenid-style filter feeder. For example, *Fang et al. (2023)* considered fluting along the lateral margin of the premaxilla (*Motani et al., 2015*) as evidence for the presence of baleen. However, it is important to note that there is no hard evidence to show that they are indeed baleen impressions. The fluting, if any, is seen in only one specimen (WGSC V26000) while lacking in other specimens that expose the ventral side of the rostrum (IVPP V4068 and WGSC V26004—the latter is a laterally-exposed specimen but has the right premaxilla showing its ventral view). Most of the impressions in WGSC V26000 are faint, allowing different interpretations—compare fig. 3E of *Fang et al. (2023)* with fig. 1A of *Motani et al. (2015)*. Also, of the 13 impressions figured by *Fang et al.* (*2023*, fig. 3E), it is difficult to find

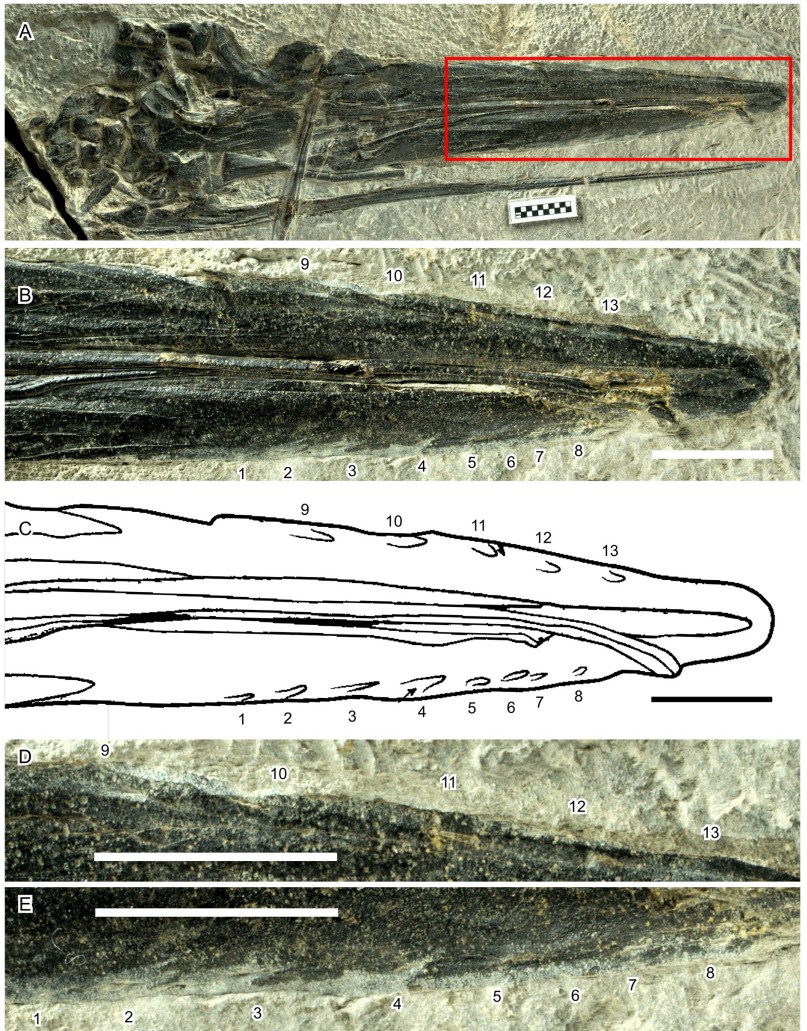

**Figure 8 Hypothetical soft-tissue impressions in *Hupehsuchus nanchangensis* (WGSC V26000).**
(A) Cranial region of the specimen. (B) Close-up of the area marked by a red box in (A). (C) Tracing of fig. 3E of *Fang et al. (2023)*, showing the same area as in (B). (D) Further close-up of the right premaxillary margin. (E) Further close-up of the left premaxillary margin. Numbers mark the position of the impressions. Their relative arrangement is kept constant across (B)–(E). Scale bar is 10 mm.

corresponding bone features for the five on the right premaxilla (Fig. 8, positions 9–13), as well as half of them on the left (Fig. 8, positions 5–8), which may be parts of a continuous groove. This groove may continue as posteriorly as position 3. Positions 1–4 stand out because of color change during preparation—the bones are dark except along the jaw margin where they appear light gray as a result of preparation. This, however, suggests potential biases from inconsistent preparation. Thus, the presence of impressions is controversial at best.

Also, there is a fundamental difference in the placement of the fluting between *Hupehsuchus nanchangensis* and baleen whales. The equivalent fluting structure are the deep palatal sulci evident on the maxillae of living mysticetes which is associated with the

neurovasculature that enables baleen to grow and expand in the intraoral space (see more below). In *Hupehsuchus*, the placement of fluting intersecting the lateral margins of the premaxilla, even if it was indeed scarring from a soft tissue structure, would suggest that such a structure would extend laterally from the jaw margin, somewhat resembling the arrangement of teeth of *Mesosaurus* and *Ctenochasma*. The use of such an extraoral structure has been suggested but not experimentally demonstrated for filter feeding in stem mysticetes (*e.g.*, *Llanocetus*, see *Fordyce & Marx, 2018*); instead, the more appropriate analog for fluting in *Hupehsuchus* might be the so-called gum teeth of Dall's porpoises (*Phocoenoides dalli*, see *Benson, 1946*). Additionally, fluting only occurs along the anterior half of the rostrum, where the rostrum is likely wider than the mandible (Fig. 1E).

The suggestion of palatal soft tissue structure related to filter feeding in *Hupehsuchus* has direct parallels and echoes similar challenges with the recent debate about inferring baleen in stem relatives of Mysticeti. Some researchers (*Deméré, Berta & McGowen, 2005*; *Ekdale & Deméré, 2022*) have proposed that foramina present on the palates of fossil mysticetes bearing adult teeth (*e.g.*, some species of Aetiocetidae) signify the presence of baleen, potentially in an incipient form. While palatal foramina in some extant Mysticeti show deep sulci that reflect the accommodation of neurovasculature to support the base (or Zwischensubstanz) of the baleen plates (*Pinto & Shadwick, 2013*), this exact condition is not patent in all extant lineages (*Peredo et al., 2018*). Developmental data from living mysticetes indicate that the location for baleen plate growth is also where incipient tooth buds are resorbed, hinting at a shared co-option of gene expression between tooth and baleen growth (*Thewissen et al., 2017*). While out- and ingroup comparisons are limited for *Hupehsuchus*, along with the lack of extant descendants, the takeaway lessons from the ongoing debate about soft-tissue filter feeding structures in extinct mysticetes would suggest that it is not a simple or clear inference for such structures in ichthyosauromorphs (nor other marine reptile lineages).

*Fang et al. (2023)* also suggested that the loose articulation of the rostrum in *Hupehsuchus nanchangensis* was an indication of filter feeding, citing *Berta et al. (2016*: 1272) who listed loose articulation of rostral bones to the braincase and with each other as one of the four shared features of baleen whales. *Berta et al. (2016)* stated that the features would "help to either filter food from water (*e.g.*, baleen), or to expand the oral cavity to facilitate an intake of large volumes of water." The link between this feature and filter-feeding mechanisms rests on whether the feature allows expansion of the oral cavity by, *e.g.*, raising the rostrum relative to the braincase. However, there is no evidence to suggest *H. nanchangensis* could lift the rostrum relative to the braincase or expand the rostrum in life. Its rostrum was loosely articulated to the braincase, as in other ichthyosauromorphs and many reptiles, but not necessarily to the dermal skeleton above the braincase. Also, the rostrum of *H. nanchangensis* was dorsoventrally shallow as in other ichthyosauromorphs but unlike in baleen whales. Slight dorsal bending of the rostrum would not probably allow the volume of the oral cavity to increase sufficiently for filter feeding, although a future study may explore how such flexion could create a minimal space in the oral cavity to do so.

Among extant mysticetes, the overall cranial architecture of balaenopteroids (and balaenids to a lesser degree) show clear signs of kinesis, specifically at the juncture between the rostrum and cranium (*Bouetel, 2005*). In balaenopterids (and gray whales), this plane of articulation presents the interdigitation of several elements from the rostrum (*e.g.*, maxilla, premaxilla) with the cranium (*e.g.*, frontal, mesethmoid) where the sutures are loose and hardly ankylosed (*Pivorunas, 1977*). Functionally, this kinesis allows the rostrum to flex both: (i) bilaterally along the rigid keel of the vomer, which extends anteriorly to the snout tip and posteriorly to firm articulation with the neurocranium; and (ii) dorsoventrally from the rest of the cranium, which increases the space of the oral cavity and potentially accommodates some of the forces imparted on the head during the engulfment phase of lunge-feeding in balaenopterids (*Lambertsen, Ulrich & Straley, 1995*; *Werth, 2001*). Balaenids may have comparatively attenuated forces given their filter feeding mode and the lateral restriction of their rostrum, which primarily flexes dorsoventrally. Overall, we suspect that the forces imparted on the cranium of *Hupehsuchus* for any potential engulfment phase would be substantially diminished.

*Fang et al. (2023)* dwelled on the presence of the interclural space in *Hupehsuchus nanchangensis*, although there is no clear statement of how this structure relates to the mechanism of filter feeding. Preserved specimens of some cetaceans have the tips of the rostrum separated along the midline (*e.g.*, Figs. 4C–4E, 4H, 4L) but it is unknown how the degree of separation is affected by postmortem modifications through progressive drying and breakdown of collagen. For example, there is a slight gap anteriorly in a specimen of *Balaena mysticetus* (NHMUK 1986.116, Fig. 2B) but not in another specimen (UAM 15988, Figs. 1C–1D). This gap is clearly attributable to the distortion that happens to the thin extremities of mysticete premaxillae as they lose lipids over time in museum collections, altering their morphology (N. Pyenson, 2025, personal observations).

The possible existence of an intergular pouch and its expansion through bending of elastic mandibular rami as in pelicans was previously suggested for *Hupehsuchus nanchangensis* by *Motani et al. (2015)*. *Fang et al. (2023)* cited this suggestion to support their hypothesis that the species was a balaenid-style feeder. However, while this feature might be consistent with the possibility of balaenopterid-style feeding, this is not the case for balaenid-style feeding. Use of such a pouch would result in two-way water flow unsuitable for the balaenid-style feeding (Table 2). Also, even if there was an expandable gular pouch with an elastic mandible, this does not necessitate filter-feeding, as explained by *Motani et al. (2015)*.

The major difference for the intergular space between balaenids and balaentoperids relates primarily to the presence or absence of a highly muscularized tongue (*Werth & Crompton, 2023*). *Hupehsuchus* possessed a hyoid apparatus with a narrow and extended anterior process (*Motani et al., 2015*), reminiscent of snakes and chameleons. However, while this structure may suggest a narrow and manipulable tongue, such a tongue is different from the massive tongue of balaenids that helps make water passages near the filter (*Werth, 2001*). Also, similar hyoid is known in at least another hupehsuchian, *Eretmorhipis*, for which there is no evidence of baleen-like structures (*Cheng et al., 2019*).

**Table 2 Filter feeding modes *sensu lato* and their associated features known in extant vertebrates.** Lunge feeders without specific filters, such as pelicans, are tentatively included since they filter the water with the jaw rami for food.

| | Continuous | | Episodic | | |
| --- | --- | --- | --- | --- | --- |
| | | | Isolated (lunge feeding) | | Repeated |
| Water flow | Flow through | | Tidal | | |
| Water intake by | Continuous body ram | | Lunging | | Pump (tongue, jaw) |
| Water drained by | Continuous body ram | | Pump (mandible, tongue) | | Pump (tongue, jaw) |
| Oral cavity size | Very large | Small | Expandable | | Large |
| Pharyngeal cavity size | NA | Large | NA | | NA |
| Filter | Present | Present | Present | Jaws themselves | Present |
| Relative prey size | Small | Small | Small | Large | Small |
| Examples | Balaenidae | Sardines | Balaenopteridae | Pelicans | Ducks |

What would non-mammalian filter-feeding modes look like in extinct marine tetrapods? In general, amniotes are constrained by anatomy and physiology that permit only certain methods of feeding (*Collin & Janis, 1997*). For example, gill-based filter feeding is not available because amniotes lack gills or a pharyngeal cavity, leaving the oral cavity to be the main filtering area. There are at least three types of filter feeding among extant vertebrates, grouped by continuous flow, as in balaenids and many fish, or episodic flow, as in balaenopterids and ducks, with the latter further divided into two depending on whether the behavior is isolated or repeated (Table 2). Table 2 tentatively includes lunge feeders without a specific filter, *e.g.*, pelicans (*Field et al., 2011*), because they do filter the water with the jaw rami to collect food items that are large. This designation, however, needs to be scrutinized in the future. For continuous filtering, anatomy ideally facilitates a system where water enters a large filter area from one end and exit from the other continuously. In swimming vertebrates, such exits need to be located behind the oral or pharyngeal cavity, whichever the filter area may be. *Hupehsuchus* lacked these features, so it was most likely not a continuous filter feeder. Episodic filtering may occur with isolated lunging or repeated pumping (Table 2). In this mode, anatomical system ideally allows a tidal motion of water, where the water first enters the open mouth and then the flow is reversed to exit from the mouth, passing a filter on its way out. One large gulp as in balaenopterid whales or many small pumps as in some ducks (*Crome, 1985*) and flamingos (*Jenkin, 1957*) may be used and it is necessary to have space to store the water momentarily before filtering, usually a somewhat expanded oral cavity or an expandable mandibular pouch, as well as a filter structure surrounding the temporary water storage area. *Hupehsuchus* lacked such an oral cavity (Fig. 9) but it possibly had an expandable mandibular pouch (*Motani et al., 2015*). It most likely lacked a filter as discussed above, and, even if it was present, it only paved the anterior part of the snout (Fig. 9). Therefore, of the types of feeding in Table 2, pelican-like lunge feeding is the most probable candidate.

Bony fishes and chondrichthyans also provide some clues. Filter-feeding as a feeding mechanism in bony fish has Devonian origins, with gill rakers forming a mesh-like

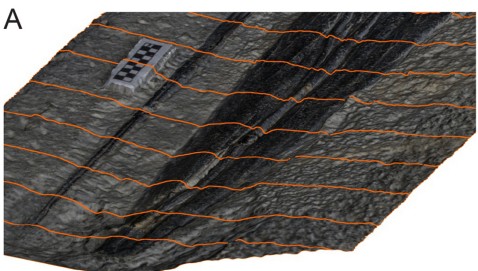
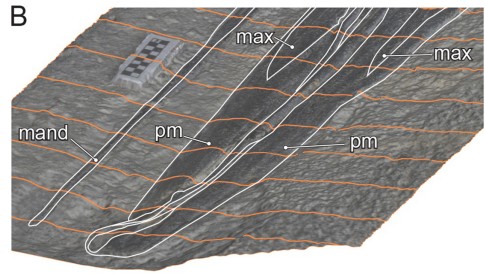

**Figure 9 Oblique view of *Hupehsuchus nanchangensis* (WGSC V26000) *via* a 3D photogrammetric reconstruction.** (A) with tracing lines for topographic relief. (B) A total of 50% opacity with elements labeled: mand, mandible; max, maxilla; pm, premaxilla. Scale bar is 10 mm.

arrangement that provided the main capture mode in the oral cavity (see *Coatham et al., 2020*). In living bony fishes, the gill rakers trap prey-laden water either by the forward motion of water streaming into the mouth (ram-feeding) or by sucking water into the oral cavity *via* rhythmic contractions of pharynx (pump-feeding; see *Sanderson & Wassersug, 1993*). The gill rakers then separate prey from water flow *via* a combination of cross-flow and dead-end filtration (*Hamann et al., 2023*). It is likely that both types of filtration are involved with other extant ram-feeding suspension feeders, such as whale sharks and baleen whales (see *Hamann et al., 2023* and *Werth & Potvin, 2024*), along with ricochet filtration in manta rays (see *Divi, Strother & Paig-Tran, 2018*).

While *Fang et al. (2023)* only considered cross-flow filtration (*i.e.*, the mechanism underlying balaenid-style filter feeding) in *Hupehsuchus*, there are clearly other filtration mechanisms that marine tetrapods possibly evolved that vary by filter type, flow regime, and particle size for filter-feeding (*Werth & Potvin, 2024*). Ichthyosauromorphs did not have gill rakers and, absent bony filters observed in Jurassic ctenochasmatid pterosaurs, the lack of a filter structure makes a filtration mechanism implausible in these marine tetrapods (*Qvarnström et al., 2019*). Teeth may provide a potential basis a filter, but neither seals nor stem mysticetes used teeth in this manner (*Hocking et al., 2017*; *Geisler, Beatty & Boessenecker, 2024*), and interlocking teeth pose several challenges for forming such a filter.

More importantly, the association of specialized organs with particular filter-feeding modes highlights an important consideration for identifying filter-feeding in extinct marine tetrapods: taxa lacking specialized organs (*i.e.*, tooth-like combs, soft-tissue filters) still meet more inclusive filter feeding categories. For example, many skim feeding seabirds lack specialized oral structures: pelicans, for example, employ lunge-feeding lacking the ventral throat and mandibular organs that rorquals possess (see also Table 2). By extension, it is entirely possible that extinct aquatic or marine tetrapods evolved specialized structures composed of soft tissues that did not adequately preserve in the fossil record.

Beyond mechanism, the body sizes of non-tetrapod filter feeders provide useful comparative insights. Extinct marine tetrapod lineages may not have been restricted to the physiological allometries that limit mammalian filter-feeding. It is possible that diapsid or

sauropterygian physiology permitted filter-feeding at smaller body sizes that mammals cannot sustain; or they were able to access densities of prey in nearshore shallow foodwebs that are physically or behaviorally inaccessible to large-bodied predators (*Cade et al., 2020*). In aquatic ecosystems paddlefish (Polyodontidae) filter-feed readily at body sizes closer to many Triassic sauropterygians, but their filter and mechanism may not be suitable analogs for tetrapods.

The ecological success of large extant filter-feeders owes primarily to the scaling efficiencies on specific densities of prey-laden water (*Goldbogen et al., 2019*). At low densities (*i.e.*, diffuse aggregations of krill or fish), the energetic return for either balaenid- or balaenopterid-style filter feeding is insufficient, and thus foraging strategies in these mysticetes optimize the availability of high-density prey aggregations in time and space (*Goldbogen et al., 2019*; *Abrahms et al., 2019*). Feeding on smaller aggregations of prey requires delayed jaw expansion in mysticetes, suggesting that prey escape behavior is decoupled from predator engulfment strategies (*Cade et al., 2020*). For Triassic ecosystems, it is unclear what prey items *Hupehsuchus nanchangensis* would ambush nor in what densities: aside from marine reptiles, other food webs components in the Nanzhang-Yuan'an fauna are poorly understood at best. This fauna assembled in an intracratonic basin in the middle of a shallow carbonate platform, isolated from the main body of the sea. This restricted basin seems to show high richness of odd and endemic marine reptile species but with limited records of potential prey organisms (see *Chen et al., 2014*; *Motani et al., 2015*; *Cheng et al., 2019*). Bromalite fossil evidence from coprolites would provide one line of evidence for the zooplankton in the Nanchang-Yuan'an Fauna, as suggested for filter-feeding ctenochasmatid pterosaurs in the Jurassic (*Qvarnström et al., 2019*).

Lastly, it is possible other factors, aside from developmental constraints and geographic restrictions, prevented marine tetrapods from evolving filter-feeding during the Mesozoic. By the mid to late Mesozoic, filter-feeding pachycormid bony fish reached baleen-whale body sizes of several meters in length and were distributed globally. *Friedman et al. (2010)* suggested that the ecological occupancy of pachycormids for 100 million years effectively excluded marine tetrapods from exploiting this trophic strategy. Filter-feeding has evolved repeatedly in vertebrate evolution since the Devonian (*Coatham et al., 2020*), arguably related to increases in ocean primary productivity (see also *Pyenson & Vermeij, 2016*). While filter-feeding evolved only once in Cenozoic mammals, we cannot exclude the possibility that other marine tetrapods evolved filter-feeding in the Mesozoic, even if *Hupehsuchus* does not appear to qualify. Primary productivity provides perhaps the single most important determinant for the evolution of filter-feeding in marine tetrapods; the timing and context of productivity rises should be examined, along with other possible factors, for marine tetrapods in the Mesozoic as well.

## CONCLUSIONS

*Fang et al. (2023)* argued that *Hupehsuchus nanchangensis* represented a lone example of a filter feeding tetrapod in the Early Triassic, over 225 million years prior to the likely origin of filter feeding in whales (*Peredo et al., 2018*). We outlined a set of approaches to validate the quantitative reasoning for *Fang et al.*'s *(2023)* conclusions, which appear erroneous

and, in part, difficult to reliably replicate. We also attempted a re-analysis of partitioned datasets with *Fang et al.*'s *(2023)* intent in mind where we found that that *Hupehsuchus nanchangensis* is located outside of the feeding morphospace of all living cetaceans. Equally, we underscore that *Hupehsuchus nanchangensis* lacked the intraoral space for the baleen (Fig. 1), and its body size was too small for sustaining the energetic balance of cetacean-style filter feeding (see Introduction). We therefore conclude that there is no evidence to support a filter feeding interpretation for *H. nanchangensis*.

*Hupehsuchus nanchangensis* does show, however, unusual feeding morphologies most closely associated with pelicans (*Motani et al., 2015*). Because lunge-feeding may occur in predators without filter organs, we argue that lunge-feeding is an aquatic feeding strategy more broadly employed than just in rorqual whales (*i.e.*, lunge-feeding extends beyond balaenopterid-style feeding). We maintain that *Hupehsuchus nanchangensis* shows no similarities to balaenids and dispute the implication that it was a balaenid-style feeder. In the parallel with lunge feeding, we highlight skim feeding also has more inclusive modes of feeding (*e.g.*, skimming by birds, which do not use a filter). Lastly, we note that the challenges of inferring filter feeding in extinct tetrapods strongly constrains the range of testable hypotheses for understanding whether bulk filter-feeding evolved prior to the Neogene. Lastly, our reexamination of *Hupehsuchus nanchangen* permits the first steps to identify the instrinic and extrinsic factors for filter-feeding in non-mammalian marine tetrapods. We encourage robust experimental and modeling approaches (*e.g.*, see *Werth, 2001*; *Goldbogen et al., 2017*) as an avenue for future research.

## INSTITUTIONAL ABBREVIATIONS

**IVPP**    Institute of Vertebrate Paleontology and Paleoanthropology, Beijing, China
**NHMUK**  Natural History Museum, London, United Kingdom
**UAM**     University of Alaska Museum, Fairbanks, United States of America
**WCGS**    Wuhan Centre of China Geological Survey, Wuhan, China

## ACKNOWLEDGEMENTS

We thank Geerat Vermeij for reading the original manuscript and giving suggestions. We also thank Dirley Cortés, Cheng Long, and especially an anonymous reviewer for comments and suggestions that improved the manuscript and subsequent revisions. We also thank those who have made the 3D models used in this study available, including those facilitated their availability through web sites, such as morphosource.org, phenome10k.org, and sketchfab.com.

### Funding

The authors received no funding for this work..

### Competing Interests

Nicholas D. Pyenson is an Academic Editor at PeerJ.

## Author Contributions

- Ryosuke Motani conceived and designed the experiments, performed the experiments, analyzed the data, prepared figures and/or tables, authored or reviewed drafts of the article, and approved the final draft.
- Nicholas D. Pyenson conceived and designed the experiments, performed the experiments, analyzed the data, prepared figures and/or tables, authored or reviewed drafts of the article, and approved the final draft.
- Da-yong Jiang conceived and designed the experiments, analyzed the data, authored or reviewed drafts of the article, and approved the final draft.

## Data Availability

The data and R scripts are available at Zenodo: rmotani. (2025). rmotani/hupehsuchus: PeerJ_Review (PeerJ). Zenodo. https://doi.org/10.5281/zenodo.14908223.

## Supplemental Information

Supplemental information for this article can be found online at http://dx.doi.org/10.7717/peerj.19666#supplemental-information.

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
