# Peer review of "Was Hupehsuchus a baleen whale-style filter feeder in the Early Triassic? A re-examination of the evidence"

_PeerJ, doi:10.7717/peerj.19666_

## Round 0.1 · original submission · Major Revisions

· Academic Editor

Major Revisions

Thank you for this stimulating investigation into feeding in an unusual Triassic marine reptile. This is a thorough attempt to replicate some previous research results and also go into additional depth for this particular animal. Note the comments from the reviewers for your consideration; particular attention should be given to two major areas:

- The manuscript itself discusses baleen whale style feeding, while also talking about "filter feeding" as a more general term. Reviewer 3 notes some areas where assumptions about plausibility of filter feeding could be more deeply explored, especially for non-cetacean examples. Please incorporate these suggestions as possible within your revision. Reviewer 1 had parallel comments around comparison with additional lunge feeders, although I would agree with them that analyses such as isotopic work are well outside the scope of the current paper. Based on the work presented here, I think you make a solid case that Hupesuchus did not have cetacean-style anatomy or filter feeding. However, I am less certain (as apparently some of the reviewers) that *every* form of filter feeding is excluded. This should be explored at least briefly, perhaps in the discussion.

- Two of the reviewers express a request to adjust the wording/tone in some areas of the critique. Although I generally take a broad view as to tone in scientific papers, I absolutely agree that some of the phrasing can be interpreted as speculative or unnecessarily harsh, which distracts from the important corrections you present. Please sure to give close consideration to the wording/tone in those relevant areas when revising your manuscript.

·

Basic reporting

no comment

Experimental design

no comment

Validity of the findings

no comment

Additional comments

Thank you for this excellent contribution. I enjoyed reading this manuscript. The authors offer a critical reassessment of the hypothesis by Fang et al. (2023), which suggested that Hupehsuchus nanchangensis functioned as a filter-feeding tetrapod in the Early Triassic, drawing parallels to modern balaenid whales. The authors argue that Fang et al.’ conclusions were based on flawed morphometric analyses and lack robust evidence for filter feeding in Hupehsuchus.

The authors demonstrate that the original dataset published by Fang et al. (2023) presents low-quality landmarks that are partially nonreplicable by proposing new morphospace analyses using more accurate landmark data. The authors present a more rigorous set of landmarks, reinterpret the anatomy of some specimens, and incorporated 3D data to test the filter-feeding capability of Hypehsuchus.

The manuscript could benefit from a comparative analysis with other known lunge-feeders to further strengthen this hypothesis, although this is not essential at this stage. Additionally, other functional adaptations in Hupehsuchus might be relevant to its feeding ecology. For example, integrating additional lines of evidence, such as isotopic or any other additional published ecological data, could strengthen the lines of evidence, but this is ultimately at the discretion of the authors to add these.

Overall, this work is significant and should be accepted for publication. I find the methods, data interpretation, use of 2D and 3D morphometrics, and figures to be sufficient for publication in PeerJ.

I am looking forward to seeing this great work published soon!

·

Basic reporting

The paper is a comment on a paper we published (Fang et al. 2023) in BMC Ecology & Evolution, rejecting our suggestion that the hupehsuchid Hupehsuchus nanchangensis was a filter feeder based on comparisons with modern filter-feeding whales. We are grateful for the chance to review the paper, but also for the measured style used by the authors in their criticisms.

The research revised the selection of the species and landmark points, leading to different conclusions, compared to previous study of Fang et al. (2023). The new landmark points are homologous and derived from 3D samples with greater accuracy, making the results more convincing. However, it is important to avoid emotional statements in the writing.

The main argument is that our morphometric comparison with modern cetaceans is flawed, especially by the use of incorrect measurements of the relative widths of certain skulls, making them artificially too narrow. Our response is to accept that some of our measurements could have been improved, and we accept the new morphospace analysis in which odontocete and mysticete whales are separated, and that Hupehsuchus overlaps with neither. However, this study does not reject the specific anatomical evidence we also cited for filter feeding in Hupehsuchus, namely its extremely narrow snout, the unfused upper jaw, the specialized intermediate space in the divided premaxilla, and grooves around the labial margin, all of which suggest an enlarged buccal cavity. Further, teeth are absent, and unusual feature for any Triassic marine reptile, and also supporting the idea that it used some other means of acquiring prey.

I do admire that Motani et al. ran the landmark based morphspace analyses of extinct Early Triassic marine reptile with modern marine animals. However, their arguments distorted the fact that the analyses in Fang et al. (2023) is replicable. As the authors' demonstration in line 381-382: "Therefore, the published dataset can reproduce the published results using the publishd methods", the results in Fang et al. (2023) are not "unattributatble nor replicable" (lines 31-32) nor "irreproducible" (line 642). So, statements like "artifact of multiple errors"(line 31), "unattributatble nor replicable" (lines 31-32), "irreproducible" (line 642) and few others should be deleted. Since the coding mothods in this MS is different with Fang et al. (2023), it's not surprise that different results came out. In addition, since there is no availble 3D reconstructions on the skulls of any Early Triassic marine reptiles, the 3D landmark based method (lines 260-271) is very questionable.

Therefore, we make these suggestions:

Lines 31–32, 642: There is a major inaccuracy in your wording here. You state that our results are neither "unattributable nor replicable" (lines 31-32) and that they are "irreproducible" (line 642). You say our grhs are an "artifact of multiple errors"(line 31). And yet you demonstrate at lines 381-382: "Therefore, the published dataset can reproduce the published results using the published methods". We ask that you remove the unfair implications of the words at lines 31–32 and 642. Please don’t accuse us of things that are untrue. You have plenty of valid criticisms.

Lines 30-33: "We show that....respective species" must be deleted.

Lines 60-73: need references for every sentences in this paragraph.

Lines 75-78: need references.

Lines 105: ‘largely based’ – we would suggest our inference of filter feeing was based on the morphometric analysis of course, but also on the anatomical features just noted, and described in detail in our paper. Therefore, refuting our morphometric analysis does not necessarily disprove our case, which also marshals anatomical evidence.

Lines 111: ‘lacks the intraoral space’ – this requires evidence. Of course, as you say, a small skull would have a small intraoral space, but that is proportional to the animal; nonetheless, the extremely narrow snout, splayed nasals, narrow jaws, and absence of teeth provide space in proportion and potentially equivalent to that in a mysticete.

Lines 110-122: delete or replace in Discussion

Line 165: “WGSC V26005”
This specimen is numbered as “WGSC V26004” and the correct WGSC V26005 is a Parahupehsuchus. See the paper A Carapace-Like Bony ‘Body Tube’ in an Early Triassic Marine Reptile and the Onset of Marine Tetrapod Predation.

Lines 260–271: The authors recommend the use of published 3D skulls of modern whales, and that would indeed be wonderful – but of course they know why neither they nor we do that as there are no trustworthy 3D models of the hupehsuchian skulls, so this should be perhaps reworded as more of a hope for the future rather than an implied criticism of us.

Lines 432: We are grateful for the thoroughness and care taken by the authors in checking and replotting their morphospaces. The narrow band occupied by the two hupehsuchians no longer overlaps with the morphospace area occupied by mysticete whales, as we had found. In fact, our analysis showed overlaps of all three clades. Now, with re-measured landmarks, the current authors separate all three morphospaces, which certainly makes great sense, and clarifies the substantial differences in skull shape and feeding mode of the two whale suborders. It is important to note that, with the separation they find, the hupehsuchian morphospace remains very close to that occupied by the filter-feeding mysticetes, and indeed had we and they used a larger sample of hupehuchians (only two) the convex hulls of the morphospaces of hupehsuchians and mysticetes might very well have overlapped (Fig. 7G).

Lines 433-436: the statement here is contradicted with the contents in lines 381-382.

Lines 449–457: We regret the tone of this section. Elsewhere, the authors are properly scientific and neutral in tone, and we fully accept the errors by us in our measurements. We refute the hints of malpractice and the expression, ‘cast serious doubts on the authenticity of the data.’ The explanation is simple, and surely occurred to the authors: the work is part of a student project, and students sometimes make honest mistakes as they learn. We regret the senior authors had not double-checked everything. To even hint at a lack of authenticity (= fakery) is we feel un-necessary and cruel. In fact, you follow (460–461) with a correct point about our poor practice, that measurements in some cases were taken from published rather sketchy diagrams that were not adequate for the purpose. We would appreciate rewording of the preceding remarks.

Lines 463: Is it puzzling why we did not use the lateral view? In fact, why didn’t you? We could have from the modern taxa, but the hupehsuchian skulls are flattened, and so lateral views of those risked being inaccurate and so we did not do any lateral view work – we wish we could have.

Lines 463-467: yes, why did we not use the 3D online skulls that were available by 2022. Yes, we should have, but I’m sure you can imagine why… the student project was completed before 2022. No great mystery, but poor practice by us. Please state objectively the samples in this study are more accurate than those of Fang et al. (2023), avoiding the use of emotional language.

Lines 482–501: You are repeating yourself – maybe delete this paragraph. We would appreciate if you continued the careful language and more generous tone of earlier parts of the paper. Here, in the Discussion, you seem to be determined to shred us, and this is especially cruel when the lead author is a young student, starting his career. We cannot insist of course, but we ask the authors to consider whether they have to hammer home these points about poor practice quite so often, and with repetitions, and in such terms of heavy censure.

Figure 1: It seems that the labels are unmatched with figures.

Figure 7: The caption does not explain parts E–H of the image.

Experimental design

no comment

Validity of the findings

no comment

Reviewer 3 ·

Basic reporting

Summary: The authors reexamine H. nanchangensis using similar methods to a previous study and show that there were inherent flaws in the original study, including irreproducible data and potentially flawed methodology. The authors do a nice anatomical reexamination, properly re-landmarking the skull (and in some cased re-identifying the bones) and running updated analyses. However, the tone of the overall paper borders on unprofessional. The reviewer notes that it is certainly frustrating to re-examine material only to discover that landmarks that were reportedly marked were in fact missing (ex. the nasals for K. breviceps and K. sima in the Fang et al. 2023 paper); however, the overall aggressive tone detracts from the findings from this paper. By no means is this reviewer suggesting that you should not articulate your disagreement, but your audience should come away from this paper focused on the findings from the re-examination of the data presented rather than continually remarking about how your group is particularly baffled by the previous Fang et al 2023 paper. With that said, some of the arguments herein that “support” the author’s conclusions that H. nanchangensis is not a filter-feeder are also fundamentally flawed and predicated on the assumption that ichthyosaur filtering would exactly match modern cetacean filter feeding.

Specific points of focus:
Lines 110-122: The authors set up a scenario where they have called into question whether filter feeding occurs in H. nanchangensis based off the lack of intraoral space – assuming that baleen would not be present. This assumption is based on the baleen or baleen-like filter looking similar to modern extant whales. However, one has only to look at the small intraoral space of small-sized filter-feeding fishes to counterpoint this argument, especially if H. nanchangensis is engaging in continuous ram filtration using a cross-flow filtration mechanism (ex. sardines). Indeed, the previous work by Fang suggests a balenid style filter (aka cross-flow filter), not an engulfment style filter (used by balaenopterids) that requires an enlargement of the intraoral space to hold water prior to filtration. The authors argue that the energetics of filter feeding in whales is more efficient at larger sizes, but this is again, constrained to the largest lunge-feeding balaenopterid whales. Filter-feeding as a feeding mechanism was already present since the Devonian, and so it seems unlikely that the filter apparatus would exactly match the baleen found in modern whales – a product of millions of years of evolution between ichthyosaurs and the baleen present in the oligocene.
Is it possible that these ichthyosaurs filtered more like a Pachycormiforms or even a modern day paddlefish (also in a similar size range at tens of Kg’s) rather than a lung-feeding whale? If not, why? The entire argument for/against filtering within this manuscript compare the specimen to extant whales, but this is fundamentally not a whale. In fact, when your study removed the unrealistic landmarks from the previous study, the placement of the specimen shifted from seemingly nesting within the cetacea. So, the inclusion of other medium-large filter-feeder morphology will strengthen your arguments for or against filter-feeding as a mechanism used by H. nanchangensis.

Specific issues to address:
• Figure 1: Please provide some reference for what the scale bar is in Figure 1H, I, J.
• Figure 1: What is the extra line in figure 1E above the left mandible?
• Figure 2: I am not following what the authors mean when they say “from the strictly dorsal projection”. I think they are trying to get at the image is in a normal view looking dorsally. The figure legend in 2A&B say this is a “orthographic projection from the strictly dorsal direction” – but B appears to be from a ventral view. Is this incorrect? The same goes for C&D that appears to be from both the dorsal (C) and ventral (D) view. I believe this is simply an oversight with their labeling.
• Figure 2G-J, can you please specify if the tilted angle is tilted based off the yaw or roll angle here? You say in line 222 it is never adjusted for pitch angle, yet it appears like this is indeed a rostral-caudal tilt. Adding this in for clarity would be extremely helpful.
• Figure 8: please provide data for scalebar used.
Line 566: The authors make a broad statement that slight dorsal bending of the rostrum would not allow the volume of the oral cavity to increase sufficiently for filter feeding, but as far as this reviewer knows, the minimum expansion necessary to induce filter feeding in a cetacean has not been fully explored. So, this comes off as a hand-wavey statement that is not backed up by experimental or computational data. If this reviewer is mistaken and these data do exist, please provide the citation here.

Lines 613-619: I find these arguments need to be better fleshed out. If the fossil record in the Nanchang-Yuan’an is scarce, then I do not understand the argument that the Fang 2023 paper must be incorrect and that this paper’s arguments are correct. Just because there is an incomprehensive fossil record in this region, does not mean that there was not a diversity of invertebrates and fishes at this time period. The entire paragraph neither provides evidence for or against either author’s points. The paragraph should either be scrapped, or the authors should provide more concrete evidence supporting their claims.

Lines 621-635: I am perplexed why the authors argue that these animals could not possibly be filter feeding based off the extreme gigantism of modern cetaceans. Yes, these giant filter-feeders must meet a minimum energetic demand to support their body functions and counter the heavy cost of lunge-feeding, but this does not mean that smaller filter feeders cannot exist and/or thrive. For example, extant megamouth sharks are likely engulfment feeders (or maybe ram filter-feeders) and are about the same size range as H. nanchangensis, yet they still manage to meet this energetic demand – something the present paper argues is not plausible for a ram or engulfment filter feeder. The authors’ argument here relies on the presumption that filter-feeding could not possibly function in ichthyosaurs that were much smaller critters than whales – but even when examining extant forms we see this argument is fundamentally flawed. Furthermore, Acanthodians and Placoderms were already present and presumably filter-feeding in the Devonian, well before H. nanchangensis.
The assumption that there would not have been enough zooplankton or rather not enough “patchy aggregations of zooplankton” is also flawed. Arthropod zooplankton were present since the Cambrian. I am not following this logic that because one fossil record at Nanchang-Yuan does not show a diversity of planktonic fishes and arthropods, then none must have existed. The authors cite literature about densities of plankton aggregations as a means to disprove the Fang assumptions; however, is there evidence backing up their statement that there were indeed low plankton aggregations at this time period? Counteracting a straw argument with another straw argument is not effective here. Unless the authors have assessed other records from global seas and have come to the conclusion that zooplankton were simply not present in this time period to support medium sized filter-feeders, then I do not see the relevance of this paragraph.

Experimental design

The authors re-examine the previous conclusion by Fang et al. 2023 and conclude the the initial experimental design was flawed. The authors did an excellent job with the re-examination and provide compelling data indicating that the previous paper had results that were not repeatable and in some cases, included data where data did not exist (ex. no intranasal landmarks).

Validity of the findings

The re-examination is sound and the authors used the appropriate landmarking to reanalyze the data. However the authors need to restructure the discussion (see specific notes).

Additional comments

I urge the authors to include other comparisons of filter feeders especially those in similar time periods before concluding that H. nanchangensis is not a filter-feeder. The paper heavily relies on comparisons to extant whales, but the modern whales filter in vastly different ways to other medium sized aquatic filter feeders - and the balaenids filter very differently from the rorquals. The presumption that an ichthyosaur (whale precursor) must filter in the same way as a modern whale is in itself a likely flawed argument.

---

## Round 0.2 · Minor Revisions

· Academic Editor

Minor Revisions

Thank you for your close attention to the comments from the previous rounds of review; the manuscript has been drastically improved, both in content and tone. I asked the anonymous Reviewer 3 to review the revised version, and they had minimal comments (see attached). Please incorporate them at your discretion.

One item to clarify in the Introduction: "and the original taxon sample contains at least one non-existent species" -- I was trying to find a later reference to this non-existent species in the text (esp. the "Assessment of the published taxon list") but couldn't find it. I may have missed it, but in either case, the issue should be clarified (is it a species that is synonymized? mislabeled? misspelled? It's unclear to me), especially in the appropriate part of the methods.

Once this is completed, I should be able to approve in short order.

Reviewer 3 ·

Basic reporting

Previously reviewer 3:
The authors did an excellent job responding and addressing comments I provided in my earlier review. I would like to particularly comment them for addressing their tone throughout and for the inclusion of the filter-feeding review which greatly strengthens the scope of their arguments.

Beyond suggesting adding a few additional filtration references to their arguments: For example, beyond citing Hamann for whale sharks, you should be sure to cite Motta et al. 2010. who originally suggested cross-flow filtration in whale sharks - though this has yet to be shown either experimentally via models or in vivo.

For bony fish filtration, two recent reviews were published and should be cited in support of your statements:

Sanderson, S. L. (2024). Particle separation mechanisms in suspension-feeding fishes: key questions and future directions. Frontiers in Marine Science, 11, 1331164.

and

Kahane-Rapport, S. R., & Paig-Tran, E. M. (2024). Suspension feeding in fishes. in Encyclopedia of Fish Physiology.

Beyond these minor adjustments, I believe the authors have successfully adjusted any issues I identified and this is ready for publishing. Excellent job.

Experimental design

No new data necessary

Validity of the findings

The authors have successfully argued their case

Additional comments

See above

---

## Round 0.3 · Minor Revisions

· Academic Editor

Minor Revisions

Thank you for your patience - I was requested to do one more round of review with one of the original authors. They have a minor comment to consider for revision; please incorporate this as appropriate. Note that the manuscript is good to go otherwise.

·

Basic reporting

The authors have revised based on comments from reviewers and the manuscript is significantly improved compared to the previous version. The authors re-examined the similarity in skull morphology between Hupehsuchus nanchangensis and extant cetaceans, using 2D and 3D morphometric analysis with more precise landmarks. Additionally, they refined the model of filter-feeding in marine tetrapods, indicating that the lunge feeding is more suitable for H. nanchangensis. We are pleased to note the removal of emotionally charged language in the revised manuscript. We accept the authors' perspective and agree that the revised version is suitable for publication.

However, there is a minor issue in the manuscript that needs to be considered by the authors. In line 537, the authors state the opinion that the fluting structure is hard to evidence the baleen in Hupehsuchus nanchangensis. But it is noted that the article of Peredo, Pyenson & Uhen, 2022 was retracted, which is unsuitable to refer the article. It would be better for the authors to cite other papers to support their arguments.

Experimental design

no comment

Validity of the findings

no comment

Additional comments

no comment

---

## Round 0.4 · accepted · Accept

· Academic Editor

Accept

Thank you for your quick turn-around on the final round of comments; in my opinion, the manuscript is now ready for publication.